# Glaucoma Detection and Classification Using Improved U-Net Deep Learning Model

**DOI:** 10.3390/healthcare10122497

**Published:** 2022-12-09

**Authors:** Ramgopal Kashyap, Rajit Nair, Syam Machinathu Parambil Gangadharan, Miguel Botto-Tobar, Saadia Farooq, Ali Rizwan

**Affiliations:** 1Amity School of Engineering and Technology, Amity University Chhattisgarh, Raipur 493225, India; 2School of Computing Science and Engineering, VIT Bhopal University, Bhopal 466114, India; 3The Home Depot, 5508 Ivy Summit Ct, Cumming, GA 30041, USA; 4Department of Mathematics and Computer Science, Eindhoven University of Technology, 5600 MB Eindhoven, The Netherlands; 5Research Group in Artificial Intelligence and Information Technology, Department of Mathematics and Physical Sciences, University of Guayaquil, Guayaquil 090514, Ecuador; 6Department of Ophthalmology, Shifa College of Medicine (SCM), Shifa International Hospital, Islamabad 44000, Pakistan; 7Department of Industrial Engineering, Faculty of Engineering, King Abdulaziz University, Jeddah 21589, Saudi Arabia

**Keywords:** deep convolution neural network, improved U-Net, image segmentation, classification, DenseNet-201 model

## Abstract

Glaucoma is prominent in a variety of nations, with the United States and Europe being two of the most famous. Glaucoma now affects around 78 million people throughout the world (2020). By the year 2040, it is expected that there will be 111.8 million cases of glaucoma worldwide. In countries that are still building enough healthcare infrastructure to cope with glaucoma, the ailment is misdiagnosed nine times out of ten. To aid in the early diagnosis of glaucoma, the creation of a detection system is necessary. In this work, the researchers propose using a technology known as deep learning to identify and predict glaucoma before symptoms appear. The glaucoma dataset is used in this deep learning algorithm that has been proposed for analyzing glaucoma images. To get the required results when using deep learning principles for the job of segmenting the optic cup, pretrained transfer learning models are integrated with the U-Net architecture. For feature extraction, the DenseNet-201 deep convolution neural network (DCNN) is used. The DCNN approach is used to determine whether a person has glaucoma. The fundamental goal of this line of research is to recognize glaucoma in retinal fundus images, which will aid in assessing whether a patient has the condition. Because glaucoma can affect the model in both positive and negative ways, the model’s outcome might be either positive or negative. Accuracy, precision, recall, specificity, the F-measure, and the F-score are some of the metrics used in the model evaluation process. An extra comparison study is performed as part of the process of establishing whether the suggested model is accurate. The findings are compared to convolution neural network classification methods based on deep learning. When used for training, the suggested model has an accuracy of 98.82 percent and an accuracy of 96.90 percent when used for testing. All assessments show that the new paradigm that has been proposed is more successful than the one that is currently in use.

## 1. Introduction

It is critical to acquire an accurate glaucoma diagnosis as soon as possible and in a timely manner to prevent future vision loss and damage. According to the World Health Organization (WHO), 3.54 percent of 40–80-year-olds have glaucoma. Those who are impacted may lose their vision. Individuals under the age of 40 are more prone than those over the age of 80 to getting glaucoma, which affects one in every eight people [1]. Intraocular pressure that is too high is thought to cause glaucoma because it damages the blood vessels and optic nerves in the eye. Glaucoma can cause total blindness because of its effects on the optic nerves, which can cause vision loss in both eyes. This is possible if the illness affects both eyes. When this happens, it is commonly referred to as the “snake thief of sight” because of what happens next. The disease of the optic nerve of the eye is often referred to as “sneak disease” because most patients show no obvious symptoms in the early stages of this disorder. This is because, in the early stages of the disease, most patients show no symptoms. Glaucoma causes irreversible visual loss, second only to cataracts. Glaucoma causes 12 percent of blindness in the U.S. each year. By 2040, 111.8 million people between the ages of 40 and 80 would have been diagnosed with glaucoma [2]. Alzheimer’s disease has a 4.7 percent likelihood of developing in those over the age of 70, compared to a 2.4 percent risk in the general population. The term “retinal ganglion cell loss”, abbreviated as “RGC”, refers to a wide range of disorders and events that can result in RGC degeneration and death. Defective vision, a deficiency of retinal ganglion cells, has been associated with ONH and neocortical nerve fiber layer problems. Glaucoma, if left untreated, puts a person at risk of losing both peripheral and central vision. Because there is currently no therapy or cure for glaucoma, clinicians must rely on diagnostic and therapeutic procedures to keep the illness under control.

It would be impossible to create gadgets that automatically detect eye sickness without it [3]. When you have retinal fundus imaging, you may see the vitreous, macula, and retina, as well as the health of the optic nerve. A fundus camera, which is often used by ophthalmologists, was used to image the retina. A retinal image can be used to diagnose a range of eye diseases, including glaucoma. Glaucoma is the main cause of blindness globally. Glaucoma changes the ONH’s central cup area. These factors may be used to diagnose glaucoma. The optic nerve head transmits retinal images to the brain. If you look closely at the retina in Figure 1, you will notice something called a fundus. Glaucoma, a condition that damages the visual nerves, does not show any warning signs in its early stages, but it eventually leads to blindness. The earlier glaucoma is diagnosed and treated, the better the patient’s chances of keeping the vision in the damaged eye. When glaucoma reaches a specific degree of advancement, the optic cup will be excavated. The term “cup-to-disc ratio” refers to the relationship between the optical cup and the optical disc. The CDR value may be used by ophthalmologists to monitor the progression of glaucoma. To get CDR from an image of the retina, you need a segmentation method with two steps. This procedure is carried on till the CDR is received. The increasing dimming or complete loss of peripheral vision is often the most visible indicator of glaucoma development. Here is only one example: glaucoma, for example, is known as “the wise one”, even though it is also known as “the thief of eyesight” because of the harm it does to patients’ vision. If you have high intraocular pressure, it is possible that you will see a halo-like glow when you are near bright lights. Possible side effects include blindness, eye irritation, cloudy vision (especially in babies), violent vomiting, or feeling dizzy [4].

Angle-closure, primary open-angle, congenital normal tension, pseudo exfoliative traumatic, and uveitic glaucoma are all kinds of glaucoma. The forms were found to be complicated, with different racial and ethnic groups experiencing them at varying rates. Open-angle glaucoma is caused by optic nerve damage. Figure 2 shows the optic nerve head’s anatomy. Open-angle glaucoma is common [5]. Even in a relatively short portion of a drainage canal that is just slightly clogged, fluid pressure can build up over time. It is quite probable that the patient may notice a loss of central vision before any other symptoms, such as blurring of vision in the afflicted eye’s periphery, become apparent. This is something that happens on a frequent basis. Angle-closure glaucoma, also called acute glaucoma, happens when the eye’s drainage system fails completely.

Another name for this type of glaucoma is angle-closure glaucoma. When the pressure rises quickly, blindness might occur suddenly. Several factors contribute to this sickness, including the drooping iris and tiny pupil of the eye affected by the condition. When there is a strain on the trabecular meshwork of the eye, drainage channels might be forced closer to the iris. This type of damage can occur when intraocular pressure increases excessively [6]. Drainage and secretion occur at the same rate in healthy eyes. To put it another way, when the drainage canal becomes clogged, intraocular pressure rises. This increase in pressure affects the optic nerves, which carry signals to the brain and allow the brain to comprehend visual information. If the damage is not treated, the patient will eventually go blind. So, glaucoma must be found as soon as possible to keep people from going blind. Researchers seek to identify and anticipate glaucoma using deep learning technologies before it becomes a significant problem. The proposed deep learning model for glaucoma image processing makes use of the glaucoma dataset. In this design, the DenseNet-201 model, which may be thought of as a pretrained type of transfer learning, is utilized in combination with a deep convolution neural network. This model’s properties and segments are obtained from it [7]. The DCNN classification method is used to determine whether the eye in question has glaucoma. Section 2 shows related work. The proposed methodology is shown in Section 3, and Section 4 and Section 5 depict performance analysis and conclusion, respectively.

**Figure 2 healthcare-10-02497-f002:**
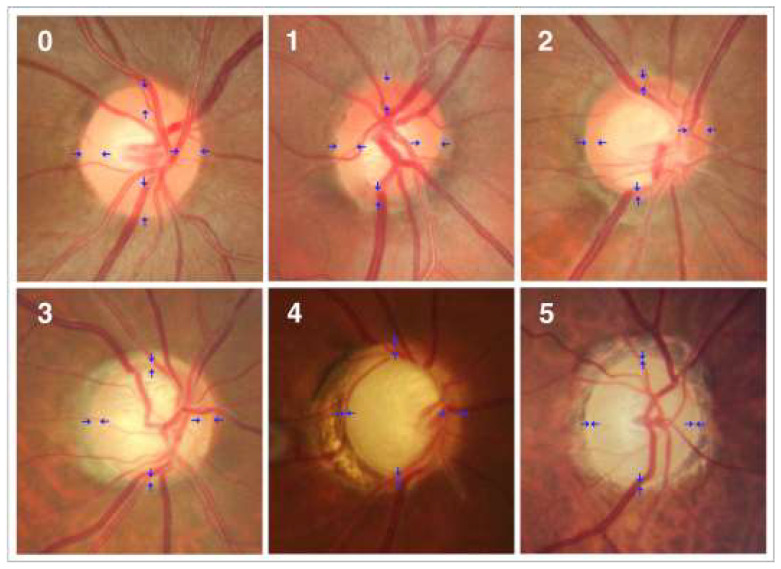
Photographs of the optic disc showing a normal disc (0) and optics with varying degrees of glaucomatous optic nerve damage (Stages 1–5). These phases are classified based on the shape of the neuroretinal rim (blue arrow) [8].

## 2. Related Work

Several writers created these models to diagnose and categorize glaucoma using a variety of research approaches and computer programs. This method resulted in the development of several study models [9,10]. Most of these models involve multiple layers of performance analysis using deep learning. We will eventually go into further depth on these topics. There are a lot of similarities between retinal disease and glaucoma, which makes them hard to tell apart. There are numerous forms of glaucoma. Some of the terms used to describe a patient’s retinal condition are shown in Figure 1. They include “macular epiretinal membrane”, “normal fundus”, “mild nonproliferative retinopathy”, “pathological myopia”, “hypertensive retinopathy”, “laser spot”, “moderate nonproliferative retinopathy”, and “mild nonproliferative retinopathy”.

Glaucoma could previously be identified with these digitally recorded retinal images. This strategy was used in the great majority of clinical investigations, making it the most frequently used methodology. Before analyzing the scan images for further study, any abnormalities detected during the discovery phase must be deleted [11,12]. During the processing stage, the blood vessels in the image were masked off and shown in such a way that the final image had no vascular features. Furthermore, an image had to be analyzed to appropriately identify the massive quantity of data that had been obtained. In this approach, the database was altered so that information could be found more quickly and readily. This was done to make the task of filtering through all the information simpler. It looks like the aim established earlier in the dialogue can be met. When it comes to defining qualities, each approach has its own specific benefit over the others. The method of image classification was used when undertaking data analysis and arranging things into categories. This project required a numerical analysis of an image to be completed correctly. To finish the study on time, the information needs to be put into different groups, like “normal” and “glaucoma”.

They used the U-Net segmentation technique to accurately identify and segment the optic cup in the retinal fundus photos they examined. As a result, a thorough investigation could be carried out. Glaucoma may be diagnosed more accurately now that the optic disc and cup have been separated. We were able to narrow down the area of interest by using an optical disc snapshot in combination with U-Net technology. An adaptive learning rate optimization strategy was used to get the most out of the training. As a result, the exercise’s good benefits were enhanced. When tested, there was a 0.15 percent mistake rate and a die coefficient rate of 984.2 percent [13,14,15]. For the first time, a large glaucoma database was used to evaluate the attention-based CNN (ACNN) model for glaucoma diagnosis. If considerable amounts of redundancy are eliminated from fundus images, the diagnosis of glaucoma may become less accurate and reliable. The ACNN model took a long time to reach a conclusion, but, in the end, it came to the following conclusion: by merging these three subnetworks, a model capable of generalizing classification, recognizing potentially dangerous circumstances, and forecasting where attention will be focused was created. It is 96.2 percent accurate and has an AUC of 0.983 percent. Few errors were made while identifying tiny trouble areas, and ROI was rarely considered [16,17]. The segmentation, distortion, form, and size of the object are all important factors to consider while using the technique. These four basic concepts served as the foundation around which the approach was built. To obtain an exact estimate of the cup-to-disc ratio, calculations must be performed. Using data from 24 separate cross-sections of the retinal fundus, we were able to distinguish between the cup and disc areas portrayed in the images. This was the first step that needed to be completed to arrive at an accurate computation of the ratio. According to the study’s findings, glaucomatous discs had an AUROC of 99.6 percent, whereas healthy discs had an AUROC of 90 percent. Using this information, the two separate sets of findings were analyzed so that they could be compared [18,19,20,21].

The split optic disc and cup should allow the system to work properly. CNN was the core component of the machine learning architecture’s deep learning capacity, which enabled automated glaucoma diagnosis. To discriminate between images of the optic disc and the fundus, researchers used neural networks trained on data from a variety of cameras. Because of the different kinds of cameras and lenses that were used to take these photos, a wide range of end products were made [22,23,24]. The model discovered that 93 percent of the 50 images properly divided the cup and 98 percent effectively separated the cup from the disc. Diaz-Pinto et al. used ImageNet-trained algorithms to diagnose glaucoma using images of the retinal fundus. This comprises Xception, as well as VGG-16 and VGG-19, and ResNet50.

Following a performance analysis, the Xception model’s claimed advantage was confirmed by testing it on five distinct datasets that were freely available to the public. Deep learning algorithms were used in the development of this model to determine if the statements were true or not. Following preliminary segmentation with the DeepLabv3+ architecture, it was determined that the optical disc region would be best encoded with many deep CNNs [25,26,27]. This finding was achieved after some preliminary segmentation. There were a lot of different learning algorithms that had to be used to find the data. These strategies comprised support vector machine (SVM)-based feature descriptor learning, transfer learning, and a combination of the two. These and other tactics were employed to achieve the aim [28,29,30,31]. Because the DeepLabv3+ and MobileNet designs could be combined, they were able to divide a hard disc. To appropriately categorize all the data, our classification approach utilized distinct algorithms for each of the five different glaucoma datasets. DeepLabv3+ and MobileNet achieved object identification and segmentation accuracy values of 99.7 percent and 99.53 percent, respectively, according to ACRIMA [12].

The model has greatly simplified the process of dividing up the regions, collecting data, and categorizing the data obtained because of using the model. To solve our problem, the Gaussian mixture model, vector generalized gradient, PCA, and SVR were employed to solve it. Deep learning improved the MNet and disc-aware ensemble network glaucoma diagnostic models. A comprehensive assessment of the patient’s eye health revealed that the patient had glaucoma. As part of the DENet, a localized portion of the optical disc was combined with a global fundus image. On the other hand, M-Net was employed to isolate the optic disc and cup. To begin with, optical discs and cups might be separated using M-Net technology. As a result, the firm was able to move to new premises [32,33,34]. This method, for example, may be used to provide a glaucoma risk estimate. Image segmentation can be done without DENet, but the method can still produce good results.

Recurrent U-Net for retinal fundus segmentation: these tasks improve accuracy. We compared and evaluated the model’s accuracy using two segmentation strategies. Patients who had both procedures had an unforgettable experience, as the retinal arteries were separated first, and then the optic cup and disc were pulled apart during the second treatment. Both operations were performed on the rear of the eyeball. Using the Drishti-GS1 dataset, the model separates the optic disc properly 99.67 percent of the time and the optic cup correctly 99.50 percent of the time [14]. Preprocessing is the first stage in the image segmentation process that must be completed before any segmentation can take place. This was done before the segmentation process began, which explains why. At this point, images taken with the optic cup and disc must be inspected. The properties of the optic disc must be established using the hybrid feature descriptions. Researchers were able to evaluate several images on optical discs using these photos. Additional features include an accelerated robust feature set, a directed gradient histogram, and a locally based binary pattern. More high-level features were revealed because of the use of CNN. Traits were categorized based on how closely they resembled or differed from one another, as well as their value in relation to the other criteria on the list. We used different multiclass classifiers, like SVM, KNN, and random forest classifiers, to look at the fundus images.

A new technique for glaucoma diagnosis involves merging the findings of two distinct types of measurements, namely temporal and geographical data. Because they conducted it this way, they were able to use data from both time and place. As a result of these characteristics, a reliable diagnosis of the eye condition was achievable (from the static structural system). Glaucoma sufferers might be identified and treated more successfully with this technology. The capacity of this classification technique to extract spatial and temporal data from fundus images is enabled using recurrent neural networks (RNN), which serve as the model’s basis. Even in the 1970s, researchers were interested in the prospect of exploiting the spatial information contained in images as a glaucoma diagnosis tool. When CNN and RNN were used together, which considered both the temporal and spatial parts of glaucoma, a better diagnosis was made. 

## 3. Proposed Methodology

Deep learning models are recommended for glaucoma diagnosis and classification. These models would be trained using retinal fundus images from the Glaucoma database. For feature extraction, Densenet-201 models that had previously been trained and prepped were used. To complete the segmentation process, the U-Net architecture is necessary. Both tasks are carried out using the DCNN architecture and the DCNN algorithm. The dataset is split down and extensively analyzed. The glaucoma dataset was used for the objectives of this inquiry [17]. In addition to the ground truth images with the extension (.jpg) that are included in the collection, the data set includes 650 color retinal fundus shots with the extension (.jpg) (.mat). The retina images were taken as part of the Singapore Malay Eye Study (SiMES). Researchers may test their computer-aided segmentation algorithms using the glaucoma database, which also makes retinal ground truth images available to the public. Following the processing of a request, the glaucoma dataset will be made accessible online for public use. After the preprocessing step was finished, the 650-image dataset was split into 162 test images and 488 training images. As shown in Figure 3, an input image is convoluted, then down-sampled, and then multiple rounds of convolution are applied to extract the features, which are then up-sampled and given to the U Net Architecture for further processing.

The optical cup segmentation tool was created using the U-Net platform’s deep learning technology. The U-Net architecture is now used for most medical image segmentation. As seen in Figure 3, U-Net segmentation utilizes a detailed architectural architecture. Deep learning is the approach of choice for maximizing the return on investment (ROI). The minor feature extraction process, illustrated in Figure 3, has a considerable influence on the segmentation outcome. When the section is finished, the optical cup will be the main structural component. In Figure 3, processed features are given to the U-Net as an input, as shown in Figure 4 and algorithm 1 explains the process, for the further feature extraction and segmentation process. During the planning stage, it was made to make sure that an algorithm could handle the ground-truth mask png image before it was divided down into its component components. The disc double (mask > 0) equation was used to generate this mask, whereas the cup double (mask > 1) equation was used to create the optic cup (OC) mask. Finally, the OD ground truth was utilized to locate the area of interest in the retinal fundus near the OC. For comparison purposes, the OC was chosen as a reference point. This was done so that we could get a better idea of where the object of interest (ROI) was.

The road narrows when you turn left and right, but it expands in the opposite direction. The two lines are linked together by numerous layers of jump connections (middle). These photos are then delivered to a layer that comes just after the growing route and oversees producing forecasts. After each three-padded convolution iteration, the filter banks of each convolution layer are given a rectified linear activation unit with the function f (z) max (0, z). Each of the two expanding and contracting directions has three convolutional blocks. The contracting strategy is followed by two layers of convolution before the maximum pooling layer. The pool is dimensions of 2 × 2 m. Prior to this layer, there was one with convolutional layers.

Box 1Explains the process, for the further feature extraction and segmentation process.1.A concatenation in the contracting route, two convolutional layers and a dropout layer in both the contracting and expanding routes, and a dropout layer, which is also called a merged layer.2.There are two steps of convolutional processing in the connecting route. In an output convolutional layer, class scores are generated pixel by pixel using a single filter and a single sigmoid activation function. There are no further tiers to complete after you have reached this point.3.Each convolution layer includes 112, 224, and 448 filters distributed over blocks 1 through 3. Blocks 5, 6, and 7 each have 224, 122, and 122 elements in their filters, which is the same number as the expanding path.4.During convolutional stacking, 448 different filters are stacked on top of each other. The proposed DCNN used several filters to help make sure it was correct and to make sure that the model was saved in the GPU’s memory.5.The model also had dropouts at various phases of its lengthy course of action. A DenseNet-201 and a CNN [21] are both included. For the sake of this investigation, we must use a DCNN model that includes DenseNet-201.6.Deep transfer learning is used by the DenseNet-201 model to recognize images of the retinal fundus and glaucoma occurrences in the input dataset.

A DenseNet-201 model that has been pretrained is used to extract dataset features, and a DCNN model is utilized to categorize the data. The resolution of the image being read is 256 by 256 pixels. Figure 5 depicts the DCNN architecture of DenseNet-201.

When the amount of data acquired is not very vast, TL may be beneficial in CNN instances. TL’s method for applications that use relatively smaller datasets takes advantage of the model learned from huge datasets like ImageNet. Because this is not required, the quantity of data required to train a deep learning algorithm is considerably reduced, allowing it to be employed for a wide range of applications. It is feasible to train one model for an assignment and then use it to train another model for a very similar job. If you want to get the most of your TL network, optimizing it rather than training it from the ground up might save you time and effort. Models that have previously been trained on large datasets can be utilized to train new models on smaller, tagged datasets. This method makes it possible to do this. Training time and computing requirements can both be lowered. TL educates models faster than starting from zero. The DenseNet-201 uses a condensed network to improve performance and build highly parametrical, effective, and simple-to-train models. DenseNet-201 has done remarkably well on datasets like ImageNet and CIFAR-100. As seen in Figure 5, DenseNet-201 offers direct connections that span from one layer to the next, enhancing connectivity. To make the application easier, it merges the feature map layers 0 through 1 into a single tensor. A transition layer is a network architecture component. Following this layer comes a 1-1 convolution layer, which is followed by a 2-2 BN pooling layer. This division is done to generate dense blocks for down-sampling. The “H” hyper-parameter sets the growth rate of DenseNet-201 and demonstrates how a dense architecture might improve performance. Despite its modest growth rate, DenseNet-201 functions well because its architecture employs feature maps to represent the overall state of the network. Therefore, the network can adapt to changing environmental conditions.

As a result, the current layer has access to all function mappings from prior layers. The number of input feature mappings at each layer, denoted by “fm”, may be calculated as follows: for each layer, (fm) I = H0 + H1, “H” feature maps relate to the global state. Input layer channels come from H0. Each 3 × 3 convolution layer contains an extra 1 × 1 convolution layer to speed up processing. This reduces their size as input feature maps are generally bigger than output. The 1 × 1 conv layer developed feature maps before it was called a “bottleneck”. Throughout the classification stage, FC layers function as classifiers. The properties of the image are used to calculate the likelihood of a segment appearing in the image. Figure 6 depicts how the DenseNet-201 design works. Activation functions and dropout layers are widely used to create nonlinearity and reduce overfitting. To categorize the data, we employed two deep layers of 128 and 64 neurons, respectively. Before utilizing DenseNet-201 for binary classification, sigmoid activation was employed. This was done to improve accuracy. Before the thick FC layer, every neuron in the brain was an FC, even if it was in a different layer. For the FC layer, we used mathematics to transform a 2D feature map into a 1D feature vector. With a specified probability and a 0–1 distribution, the Bernoulli function produces vi = 1. This random vector is generated by Bernoulli. Dropout prevents random neurons from firing in the first two FC levels. There is no overfitting from non-normalized input data; the sigmoid activation function returns 0 or 1. It seems logical to categorize images depending on their glaucoma risk. The sigmoid function is mathematically represented as follows: S = (1/(1 + e(-xz))) (1) in this equation. S represents the neuron’s output. The inputs and weights in that sequence reflect the variables xi and zi.

## 4. Results

This dataset evaluates the DCNN, U-Net, and DenseNet-201 models. When evaluating a model, its F-measure, accuracy, precision, recall, and specificity are all considered. A comparison of research adds to the proof of the strategy’s validity. Inception additional deep learning models, such as ResNet, ResNet 152v4, DenseNet-169, and VGG-19, are now being utilized to analyze CNN classification findings. Every Python test uses only one-third of the data for training, with the remaining two-thirds are used to support the performance analysis.

The primary purpose of this research endeavor is to diagnose glaucoma, as well as to assess whether individuals are impacted by the illness. The model can generate either positive or negative results depending on whether a glaucoma-infected outcome is achieved. Figure 7 shows the process of the U-Net model. With this method, you can get a rough idea of what will happen by looking at the actual positive and negative findings, as well as any other possibilities that are even a little bit likely. To put it another way, it shows the total number of correct forecasts. The FP represents the number of properly anticipated and demonstrated positive outcomes. When you see this number, you will be able to see how often you were on the right track. If the case was unsuccessful, this function counts the total number of incorrect projections. The ability of a model to correctly anticipate the behavior of a subset is known as its accuracy. Calculating efficiency is done by using it during the classification process. Its goal is to give a rough idea of how often both positive and negative classes are needed. In the classification of glaucoma fundus images, training and testing using the suggested model were demonstrated to be more accurate than with earlier models. The findings of this study are shown in Table 1 and Figure 8. When compared to previous techniques, it increased training accuracy by 1.09 percent to 3.96 percent on average. When compared to the other models being considered, the tests are 96.90 percent accurate, which is a performance gain of between 1.36 percent and 5.26 percent.

Results must be precisely predicted to be trustworthy; this also evaluates how likely a positive comment would have been received. Based on the data that are currently available, a judgement is made as to how well the results may be expected. The classification model’s number of false positive predictions has increased because of a drop in accuracy. As can be seen in Table 2 and Figure 9, the suggested model was more accurate than the models used as comparisons. The model’s training accuracy was determined to be 98.63 percent, representing an increase of 1.1 percent to 4.8 percent above competing strategies. In comparison to other models examined, the accuracy rate was 96.45 percent, implying a performance increase of 1.08 percent to 4.9 percent. Figure 8 depicts two approaches to precision analysis. In some cases, the term “recall” can also refer to “sensitivity”. The “positive predictive value ratio” calculates the percentage of total positive predictive value that is due to correct forecasts. According to the recall value of the model, false negative discoveries have had a detrimental impact on the categorization model. Memory increases when you divide the amount of information you already know and can recall by the entire amount of information you need to remember.

According to Table 3 and Table 4, this model has higher sensitivity and recall than other models. When compared to earlier techniques, recall rates increased from 1.1 percent to 4.05 percent. According to the study, the recall rate outperformed the other models by between 1.3 and 5.06 percentage points. Figure 10 and Figure 11 show the distinction.

An accurate prediction of the risk that an otherwise healthy individual would get infected with a disease is referred to as a model’s “specificity”. The total number of people whose samples were analyzed and found to be clear of the illness under consideration. Use the following equation to get a more accurate number for specificity. To establish the specificity of the test, just increase the total number of tests by one. The proposed model outperformed the other deep learning models in terms of specificity rates when compared to their total performance. The model has a 98.15 percent specificity rating compared to previous techniques, which only had specificity ratings of 0.8 to 4 percent. This is because older approaches only achieved specificity rates of 0.8–4 percent, which is the reason for the increase. According to the findings, 96.33 percent of the models tested outperformed their competitors in terms of specificity by a range of 0.6 percent to 6.4 percent. The comparison of estimated specificity values is shown in Figure 10 below.

The weighted harmonic mean of accuracy and recall, often known as the F-measure, is one method for measuring how exact an examination is. The F-measure is the more generally used term for this measurement. When establishing the validity of a computation, the distribution of the data is ignored. The F-measure is then used to ensure that the issue’s spread is accurately regulated and managed. When the dataset was used, it benefited the analysis to include categories that were uneven. Plug the following equation into your calculator to get a preliminary estimate of the F-measure. The following is the formula to calculate the value of F. The following is the F formula, which, as can be seen, asks: to what percentage of total memory does recall precision and recall itself contribute? The findings of the F-measure indicate that the recommended model performed better than the models shown in Table 5 overall. The model has already benefited from previous tactics, which increased the success rate of F-measure training from 98.50 percent to 3.7 percent. During the testing phase, it was observed that the F-measure rate of at least one model was greater by 0.08 percent to 4.47 percent than the F-measure rates of the other models. Table 5 and Figure 12 contain information that may be used to compare F-measures. The proposed model performed admirably in both the training and testing stages of this inquiry, especially when compared to the other models investigated in this study. In every regard, the model proposed was better than all preceding models. The performance of this model is close to that of the suggested model, but the overall performance of Inception’s ResNet model is bad. The first advantage of the U-Net model over previous approaches to segmentation is its ability to use data related to both global location and context at the same time. The U-Net approach has numerous other advantages. The U-Net model has several distinct advantages over other approaches to segmentation tasks. Furthermore, it improves segmentation-related task performance while requiring only a small number of training samples. This is an asset to have. We up-sample the network’s features and concatenate them with higher-resolution feature maps from the encoder network to improve representation learning through upcoming convolutions. This is the most important aspect of the U-innovation Net’s design. These maps came from the encoder network. This action will benefit the children by strengthening their representational learning. We can apply deeper networks to the process without sacrificing feature extraction performance by employing a structure known as parallel dilated convolution. This structure is used by the network model presented in this study to help deep-dilated U-Net perform better at semantic segmentation. The network model uses this structure to represent relationships.

## 5. Discussion

Glaucoma now affects around 78 million people throughout the world. By 2040, it is expected that there will be 111.8 million cases of ailments worldwide. Researchers propose using deep learning to identify and predict whether a person has the condition. Intraocular pressure that is too high is thought to cause glaucoma because it damages the blood vessels and optic nerves in the eye. The disease of the optic nerve of the eye is often referred to as “sneak disease” because most patients show no obvious symptoms. The increasing dimming or complete loss of peripheral vision is often the most visible indicator of glaucoma development. Possible side effects include blindness, eye irritation, cloudy vision, violent vomiting, or feeling dizzy. The earlier glaucoma is diagnosed and treated, the better the patient’s chances of keeping vision in the damaged eye. When there is a strain on the trabecular meshwork of the eye, drainage channels might be forced closer to the iris. This type of damage can occur when intraocular pressure increases excessively. Researchers seek to identify and anticipate glaucoma using deep learning technologies. Deep learning models are recommended for glaucoma diagnosis and classification. These models would be trained using retinal fundus images from the glaucoma database. After the preprocessing step was finished, the 650-image dataset was split into 162 test images and 488 training images. The optical cup segmentation tool was created using the U-Net platform’s deep learning technology.

The minor feature extraction process has a considerable influence on the segmentation outcome. There are no further tiers to complete after you have reached this point. A DenseNet-201 model is used to extract dataset features, and a DCNN model is utilized to categorize the data. During convolutional stacking, 448 different filters are stacked on top of each other. TL’s method for applications that use relatively smaller datasets takes advantage of the model learned from huge datasets like ImageNet. Models that have previously been trained on large datasets can be utilized to train new models on smaller, tagged datasets. Training time and computing requirements can both be lowered. Each 3 × 3 convolution layer contains extra 1 × 1 convolution layers to speed up processing. Throughout the classification stage, FC layers function as classifiers. Dropout prevents random neurons from firing in the first two FC levels. There is no overfitting from non-normalized input data. The U-Net model can generate either positive or negative results depending on whether a glaucoma-infected outcome is achieved. The ability of a model to correctly anticipate the behavior of a subset is known as its accuracy. When compared to previous techniques, it increased training accuracy by 1.09 percent to 3.96 percent on average. The classification model’s number of false positive predictions has increased because of a drop in accuracy. The suggested model was more accurate than the models used as comparisons. According to the recall value of the model, false negative discoveries have had a detrimental impact on the categorization model.

## 6. Conclusions

This paper proposes deep learning as a novel method for glaucoma diagnosis and prediction. The glaucoma dataset was utilized to train the glaucoma image analysis deep learning model. Less than a quarter of the data was used for research, with the remainder being used for classroom instruction. The data were separated using the U-Net segmentation model, and the features were extracted using DenseNet-201, a pretrained transfer learning model paired with DCNN. To detect glaucoma, the images were classified using a deep convolutional neural network. These retinal fundus images were utilized to establish whether the patient had glaucoma. After the optic cup region was recovered from the fundus photos, the resultant image data was compared to the dataset’s ground truth images. The DenseNet model was used to extract properties from segmented photos to organize the data. Previous models showed that the model was 96.90 percent accurate when tested and 98.82 percent accurate when used for training. Other models are 1.36 to 5.26 percentage points more accurate. When compared to the competition, both the comparison and the training reveal that this model produces more accurate results. The model’s conclusions can be considered accurate considering the outcomes of the performance investigation. Due to the prevalence of such findings, we were unable to produce results that were 98.4 percent correct. Soon, classifier improvements and a lower threshold are likely to correct this mismatch. These approaches can identify breast cancer, brain tumors, diabetic retinopathy, and other disorders. We document all of the measures we took in our report, as well as the lessons we learned from the subsequent evaluations. Experimental findings show that our DenseNet-201-D technique beats cutting-edge technology. Finally, we would like to see the MS classification perform better. One method is to build a variety of DenseNet-based deep neural networks and feed them a varied range of transfer learning parameters. Another area of study that might help neuroradiologists in their diagnostic work is the detection of MS-related lesions. The outcomes of this investigation may be generalizable to a variety of imaging modalities. The suggested transfer learning technique has the additional benefit of being applicable in a variety of settings, including but not limited to manufacturing, transportation engineering, industrial imaging, and others. Future attempts will include fuzzy and semi-supervised techniques.

## Figures and Tables

**Figure 1 healthcare-10-02497-f001:**
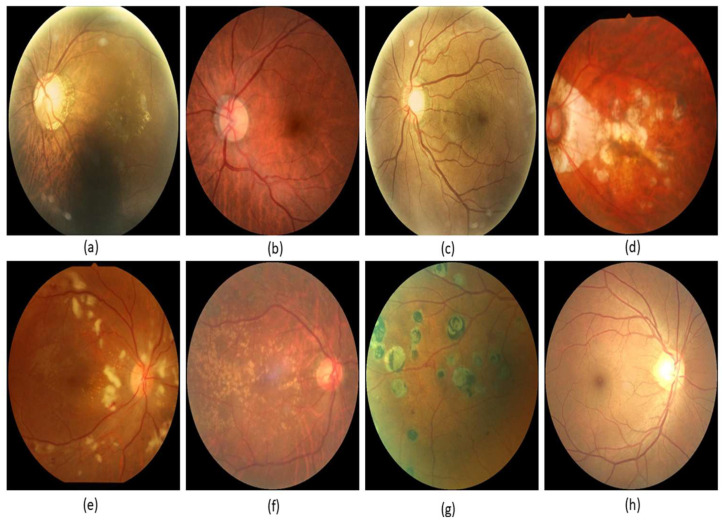
Glaucoma images: (**a**) macular epiretinal membrane, (**b**) normal fundus, (**c**) mild nonproliferative retinopathy, (**d**) pathological myopia, (**e**) hypertensive retinopathy, (**f**) laser spot, moderate nonproliferative retinopathy, (**g**) moderate nonproliferative retinopathy, laser spot, (**h**) mild nonproliferative retinopathy.

**Figure 3 healthcare-10-02497-f003:**
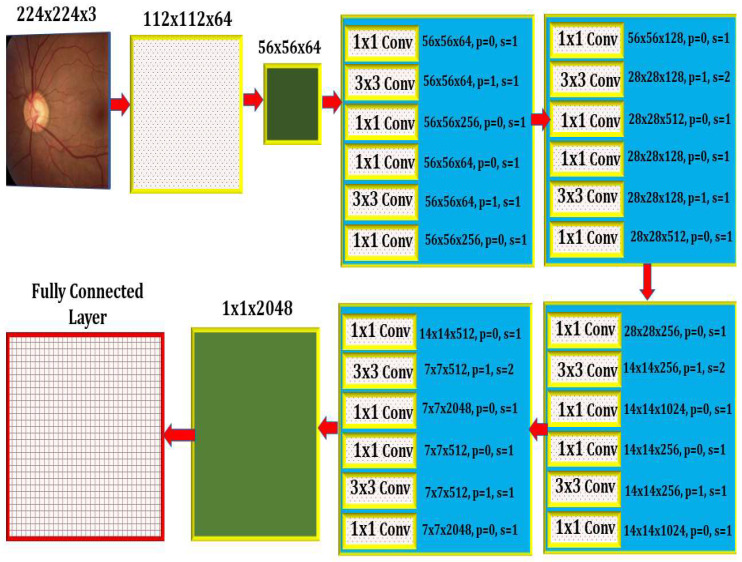
The Internal Architecture of the Proposed Model.

**Figure 4 healthcare-10-02497-f004:**
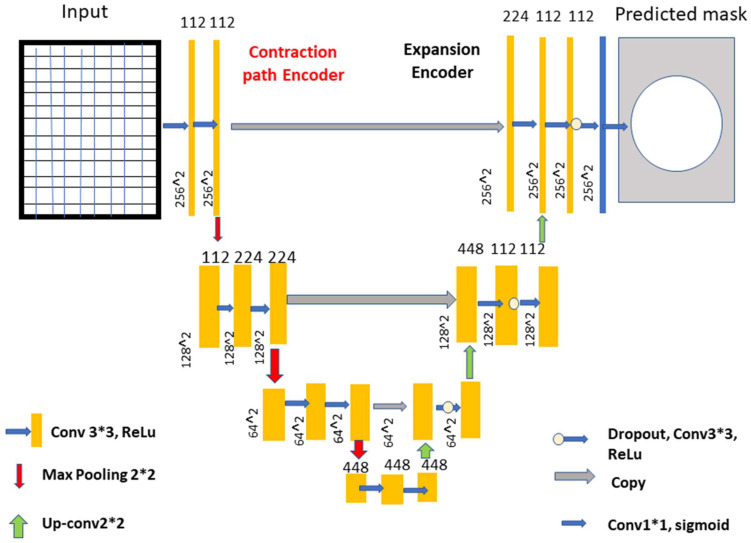
The improved U-Net model. Process see Box 1.

**Figure 5 healthcare-10-02497-f005:**
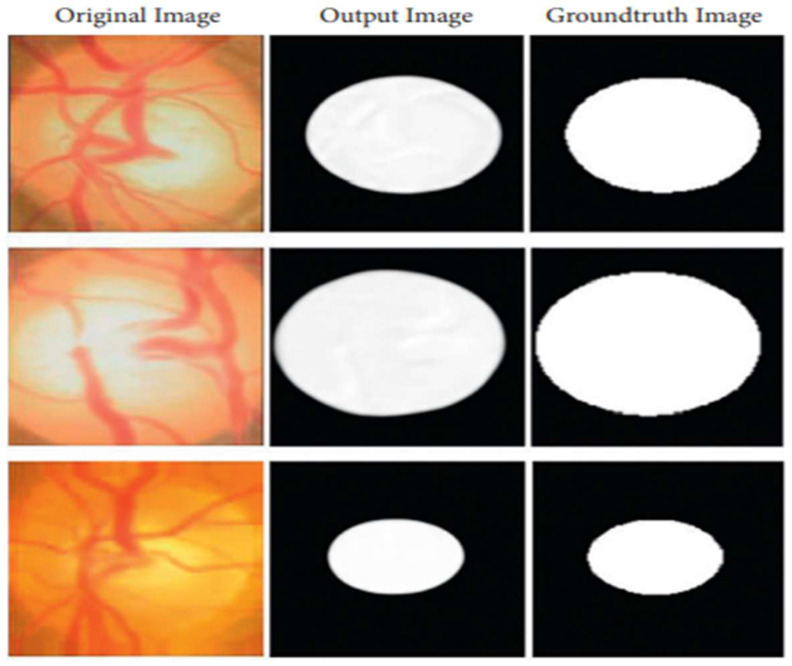
An image created by U-Net that is compared to the actual ground-truth Column 1 shows original images, Column 2 shows output images generated by the proposed method, and Column 3 shows ground-truth images.

**Figure 6 healthcare-10-02497-f006:**
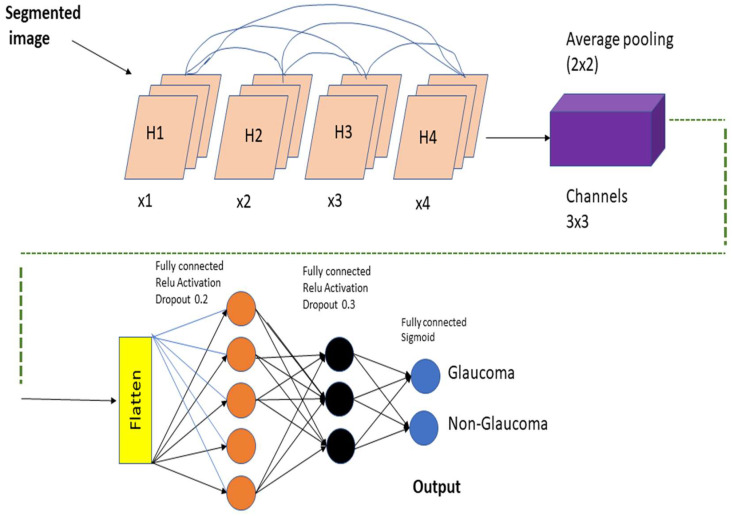
Feature extraction using pretrained DenseNet-201 model and classification using DCNN.

**Figure 7 healthcare-10-02497-f007:**
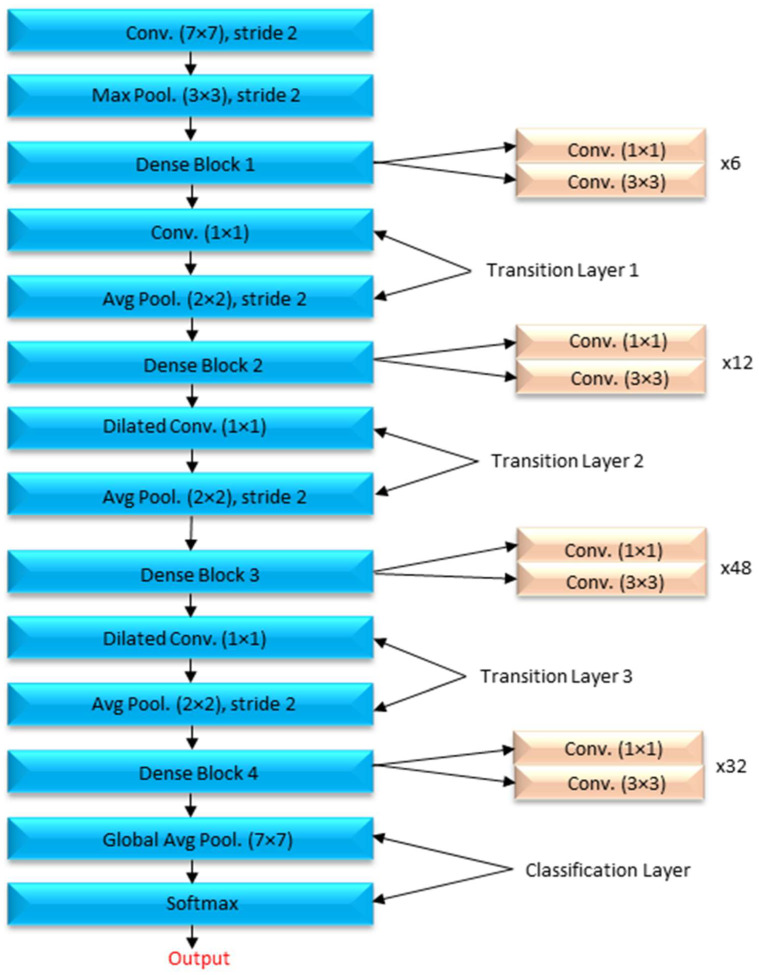
The internal process of the proposed improved U-Net model.

**Figure 8 healthcare-10-02497-f008:**
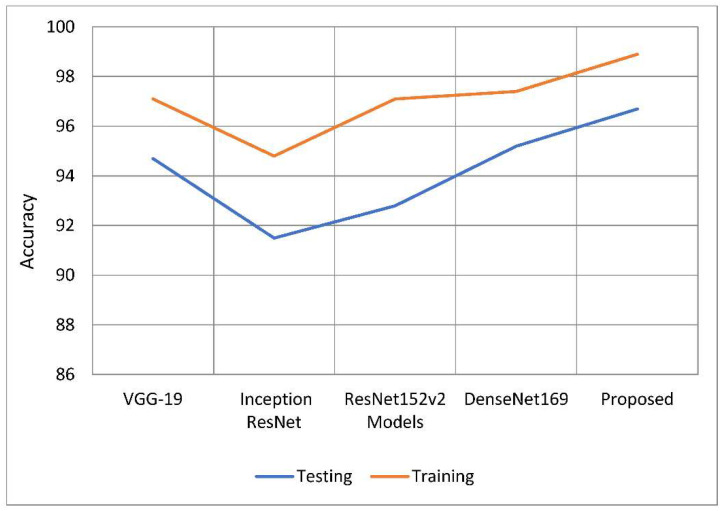
Comparative analysis of the testing and training accuracy of the proposed model with traditional models.

**Figure 9 healthcare-10-02497-f009:**
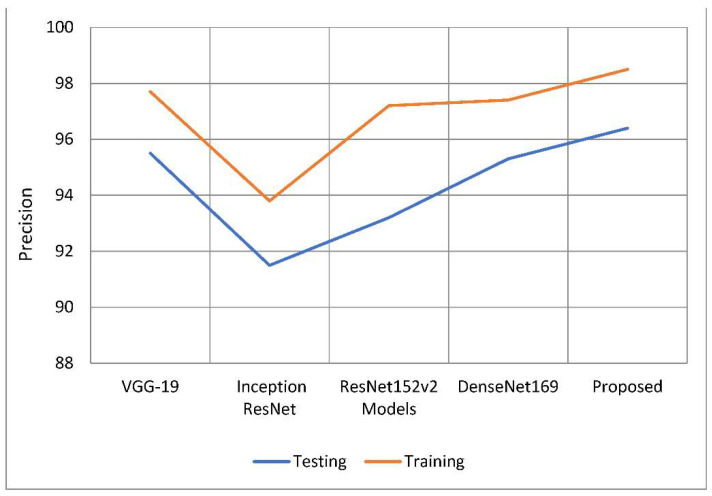
A comparative analysis of precision of the proposed model with traditional models.

**Figure 10 healthcare-10-02497-f010:**
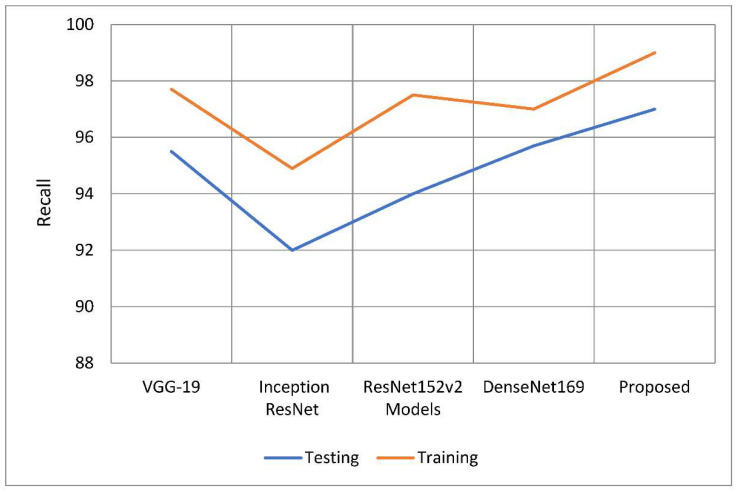
A comparative analysis of recall of the proposed model with traditional models.

**Figure 11 healthcare-10-02497-f011:**
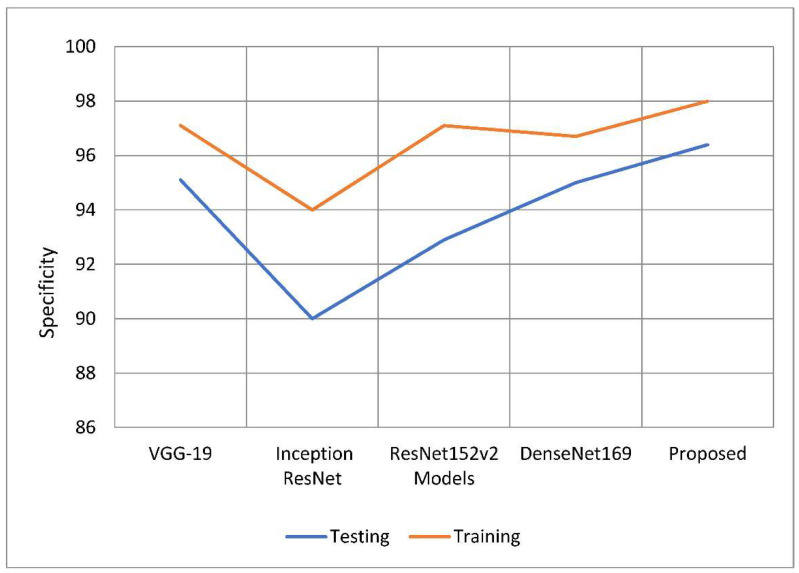
Comparative analysis of specificity of the proposed model with traditional models.

**Figure 12 healthcare-10-02497-f012:**
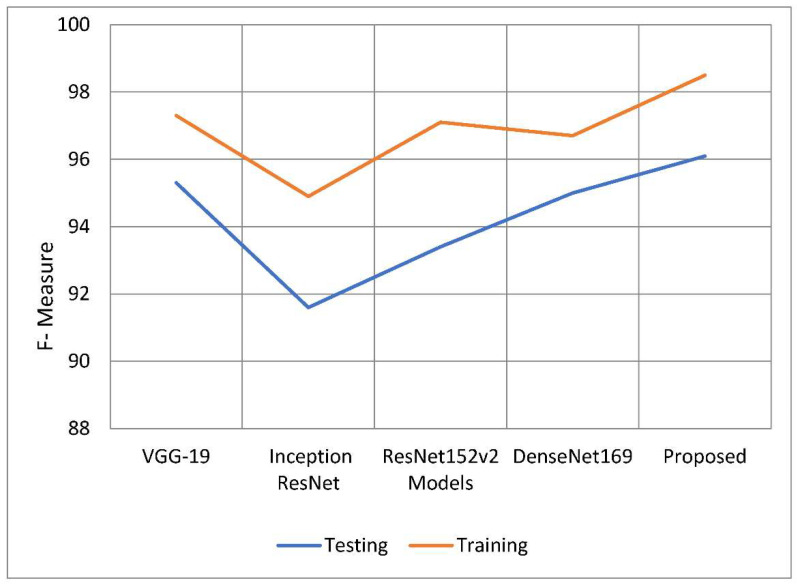
F-measure of the proposed model with traditional models.

**Table 1 healthcare-10-02497-t001:** An analysis of the effectiveness of the accuracy.

Models	Training	Testing
VGG-19	97.73	95.54
Inception ResNet	94.86	91.64
ResNet 152v2	97.56	93.21
DenseNet169	97.14	95.45
U-Net	98.11	96.37
Proposed Model	98.82	96.90

**Table 2 healthcare-10-02497-t002:** Accuracy of performance is examined.

Models	Training	Testing
VGG-19	97.30	94.70
Inception ResNet	93.81	91.52
ResNet 152v2	97.28	93.02
DenseNet169	97.49	95.37
U-Net	98.02	95.97
Proposed Model	98.63	96.45

**Table 3 healthcare-10-02497-t003:** Evaluation of the effectiveness of recall.

Models	Training	Testing
VGG-19	97.84	95.62
Inception ResNet	94.90	91.97
ResNet 152v2	97.62	94.05
DenseNet169	97.35	95.69
U-Net	97.77	96.48
Proposed	98.95	97.03

**Table 4 healthcare-10-02497-t004:** Comparison of the testing and training accuracy of the proposed model with traditional models.

Models	Training	Testing
VGG-19	97.24	95.67
Inception ResNet	94.05	89.92
ResNet 152v2	97.28	92.73
DenseNet169	97.00	94.89
U-Net	97.97	95.15
Proposed	98.15	96.33

**Table 5 healthcare-10-02497-t005:** Comparative analysis of F-measure of the proposed model with traditional models.

Models	Training	Testing
VGG-19	97.52	95.39
Inception ResNet	94.79	91.55
ResNet 152v2	97.35	93.14
DenseNet169	97.07	95.09
U-Net	97.87	96.35
Proposed	98.50	96.28

## Data Availability

The dataset can be downloaded at: “Glaucoma_dataset”, Kaggle.com, 2022. Available online: https://www.kaggle.com/datasets/sreeharims/glaucoma-dataset (accessed on 2 July 2022).

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
