# Peer review of "Glaucoma Detection and Classification Using Improved U-Net Deep Learning Model"

_healthcare, 2022, doi:10.3390/healthcare10122497_

Round 1
Reviewer 1 Report (New Reviewer)
In this paper, the improved UNet deep learning model is used for glaucoma detection and classification, and the superiority of the proposed method is verified by experiments.
Despite well organized and written, some suggestions follow to further improve manuscript:
1. Line 47 has a significantly different word spacing than the other lines, please check for changes.
2. In the result part, there are a lot of blanks on the page where the figure and table are located (such as the page in Figure 8 and Table 2), can you adjust the size and format of the chart to make the paper format more standardized and tidy
3. The manuscript lacks a discussion section. Could you please add a discussion? You may discuss your findings in comparison with the work of other groups, and discuss the novelty of your approach, and limitations.
4. Please add references to the last paragraph of the relevant work.
Author Response
Thank you for your recommendation, this comment is taken into consideration carefully and accepted in this revision. Please find the attached document.

Reviewer 2 Report (New Reviewer)
The paper proposed to identify and predict glaucoma before symptoms appear using improved U-net and CNN.
Several things the reviewer wondered about for this paper are as follows:
1. Please redraw figure 3, 4, and 6 to be one figure because these figures are belonged to proposed architecture. The advantage for redrawing is to easily understand the whole architecture of proposed system.
2. Please expand or write in detail on each caption of figure and table. Short caption makes confusing.
3. What is the advantage of improved U-Net? what is the difference between original U-Net and proposed U-Net (improved version)? Please provide the statistical report for comparison between original U-Net and proposed U-Net because it is the core idea of this paper so the reader and reviewer can believe that the proposed architecture can be useful on computer vision communities.
4.
Author Response
Thank you for your recommendation, this comment is taken into consideration carefully and accepted in this revision. Please find the attached document.

Round 2
Reviewer 2 Report (New Reviewer)
Thank you for the response. All comments from reviewer have been addressed in current version.
This manuscript is a resubmission of an earlier submission. The following is a list of the peer review reports and author responses from that submission.
Round 1
Reviewer 1 Report
The contributions of the article are not clear to me. I will suggest to summarize the main contributions and list them in the Introduction section.
A very lightweight neural network is adopted in the paper, as given in Fig. 3. I am wondering what's the motivation to design the neural network.
The related work section is not comprehensive. Many medical segmentation works are missed in the paper, such as, Quality-Aware Memory Network for Interactive Volumetric Image Segmentation.
It is very suprised to see that the model even outperforms many heavy networks like ResNet152 and DenseNet169. Could you provide more analysis regarding the results?
Reviewer 2 Report
The manuscript presents a solution to detect and classify Glaucoma based on a deep learning model. Some comments and suggestions for clarification are as follows:
#1: The figure's quality in the manuscript needs to be improved.
#2: The content of the captions of the images does not match the content presented. The author must correct it.
#3: Figure 3 needs to be explained in more detail, is the output of this process a tensor with a size 2048x1000? It will then be used for inclusion in the Unet model? Why performs this process? In the Unet model, the encoder side implements convolution layers to extract features.
#4: The author should restate the diagram and explain in detail the steps of the implementation process.
#5: Is the goal of the research detection and classification? So, what is the Unet model used in this study? Unet's output is a binary image? The input of the Densenet-201 model is the image after being segmented by Unet? Can the author clarify the meaning of Unet in the whole process?
#6: Could the author explain the difference between figure 4 and figure 7?
#7: In the results section, the authors should summarize the results on the same table as well as on a chart. This one will make it difficult and tedious for readers.
Reviewer 3 Report
The objective of this research is to describe the development of a system for automatic identification of features in retinal fundus images, characterize these features and measure their accuracy when used as image descriptors for diagnosis with artificial intelligence-based computer vision algorithms.
Medical imaging is a widely used technology for diagnostics and treatment. However, current medical image analysis methods often lack accuracy and pathological detection. Artificial intelligence-based image analysis has been widely used in recent years to solve this problem. In this paper, the authors mainly focused on spatio-temporal biomedical image analysis based on artificial intelligence techniques, especially deep learning algorithms.
This article presents an automatic temporal analysis of OCT images for glaucoma detection. They propose a method for the detection of neovascularization in ocular diseases using a deep convolutional neural network. Similar works can be found done recently in different countries with separate datasets, by doing this streamlined research the algorithm gets more educated before making any decision.
Accuracy issue:
There are certain limitations of the current method, for instance, it can’t detect wing vein. The result is not accurate and valid which affects the treatment choice and decision making. Moreover, there are many settings that need to be considered in the processing of 3D data such as voxel size, interpolation and surface smoothing.
Review:
• A good literature review was provided while doing the research. The Applied methodology part was written to the point and complete. The overall presentation of the paper was found to be good. The publication was research orientated and neutral.
• The paper was written in clear understandable English, which made it clearer and more readable.
• The research was provided with different graphs and diagrams. A better spacing in between the figures will make the figures more interpretable and the article will look more professional.
• The references on the article were found to be relevant and not older than (mostly) five years, and no excessive self-citations were found.
• The statements and conclusions on the paper were found to be coherent with similar works and the research itself.
• The deduced results were obtained after thorough mathematical calculations and are true.
Overall Recommendations:
Accept in Present Form: The paper can be accepted without any further changes.
Kind Regards,
